

# A method for generating a quasi-linear convective system suitable for observing system simulation experiments

Jonathan D. Labriola[1,2] and Louis J. Wicker[1]

[1]National Severe Storms Laboratory, Norman, Oklahoma, 73072, United States
[2]National Research Council, Washington, DC, 20001, United States
**Correspondence:** Jonathan Labriola (JDL930@gmail.com)

**Abstract.** To understand the impact of different assimilated observations on convection-allowing model forecast skill, a diverse range of observing system simulation experiment (OSSE) case studies are required (different storm modes and environments). Many previous convection-allowing OSSEs predicted the evolution of an isolated supercell generated via a warm air perturbation in a horizontally homogenous environment. This study introduces a new methodology where a quasi-linear convective system is generated in a highly-sheared and modestly unstable environment. Wind, temperature, and moisture perturbations superimposed on a horizontally homogeneous environment simulate a cold front that initiates an organized storm system that spawns multiple mesovortices. Mature boundary layer turbulence is also superimposed onto the initial environment to account for typical convective scale uncertainties.

Creating an initial forecast ensemble remains a challenge for convection-allowing OSSEs because mesoscale uncertainties are difficult to quantify and represent. The generation of the forecast ensemble is described in detail. 24 hour full-physics simulations (e.g., radiative forcing, surface friction, microphysics) initialize the forecast ensemble. The simulations assume different surface conditions to alter surface moisture and heat fluxes and modify the effects of friction. The subsequent forecast ensemble contains robust non-gaussian errors that persist until corrected by the data assimilation system. An example OSSE suggests a combination of radar and conventional (surface and soundings) observations are required to produce a skilled quasi-linear convective system forecast, which is consistent with real case studies. The OSSE framework introduced in this study will be used to understand the impact of assimilated environmental observations on forecast skill.

## 1 Introduction

Forecasts of convection can provide important guidance to forecasters ahead of an impending severe weather event; however, forecast skill is often limited because the predicted storms are sensitive to modest initial condition errors. To mitigate these errors a combination of in-situ (surface stations, radiosondes) and remotely sensed observations (e.g., radar reflectivity, satellite radiances) are blended with a prior estimate using data assimilation (Kalnay, 2002). Although skilled convection-allowing model (CAM) forecasts can be generated after assimilating commonly available observations (e.g., Johnson et al., 2013; Sobash et al., 2016; Skinner et al., 2018; Snook et al., 2019; Flora et al., 2019), much of the atmosphere at the meso- and convective scales remains unobserved. Most model state variables are also, at best, indirectly observed and thus a challenge to update





during data assimilation. These limitations introduce uncertainties into posterior estimate of the environment and degrade forecast skill. To address these concerns data assimilation experiments can be used to determine future observational networks that when assimilated, positively impact CAM forecast skill.

Rather than prematurely deploying observing systems to conduct real-world experiments, which are costly and time consuming, many studies rely upon observation system simulation experiments (OSSEs) to determine the impact of assimilating new

observation types (e.g., Snyder and Zhang, 2003; Xue et al., 2006; Jung et al., 2008; Yussouf and Stensrud, 2010; Potvin and Wicker, 2012; Sobash and Stensrud, 2013; Cintineo et al., 2016). OSSEs are simulated forecast and data assimilation experiments. These experiments assimilate simulated observations that are extracted from a nature run; a well-tuned simulation that is designed to resemble a real-world weather phenomenon (e.g., an isolated supercell, quasi-linear convective system [QLCS]). OSSEs provide an elegant strategy to test the effectiveness of different observation types, deployment strategies, and sampling

intervals because the simulated observations are easily reconfigured. To understand the impact of different observing network configurations, the data assimilation initialized forecasts are verified against the nature run. Despite the potential power of this framework, designing these experiments is nontrivial because both the nature run and model prior state must reflect complex atmospheric phenomena and uncertainties that are observed in the environment.

To ensure OSSE results are robust, it is imperative to run forecast and data assimilation experiments for a diverse range of

storm cases. Despite many different observed storm modes (e.g., Gallus et al., 2008), most convective-focused OSSEs simulate the evolution of a supercell thunderstorm (e.g., Snyder and Zhang, 2003; Zhang et al., 2004; Dowell et al., 2004; Xue et al., 2006; Caya et al., 2005; Gao and Stensrud, 2014; Kerr et al., 2015; Zhao et al., 2021). This is done in part because supercell thunderstorms produce a disproportionately large number of severe weather and tornado reports (e.g., Kain et al. 2008), and thus serve as a logical first choice. These cases are also easier to create because a realistic storm can be generated by inserting

a "warm bubble" into an unstable and highly sheared environment that is horizontally homogenous.

Only a few OSSEs simulate the evolution of non-supercellular convection such as disorganized convection (Potvin et al., 2013) or a line of storms that grows in scale (Sobash and Stensrud, 2013). To our knowledge, no idealized OSSE (i.e., experiments initialized from a sounding and a supplied mesoscale background) has simulated the evolution of a convective line initiated via a frontal boundary. Quasi-linear convective systems (QLCSs) that initiate via frontal forcing in highly sheared but

marginally unstable environments often cause severe weather in the southeastern United States during the cool season (Guyer and Dean, 2010; Sherburn and Parker, 2014; Sherburn et al., 2016). Creating OSSEs that simulate other convective initiation mechanisms (e.g., cold front, dry line boundary) and environments (e.g., high-shear, low-instability) will help to better understand how assimilated observations impact the environment and the subsequent evolution of convection.

Convective-scale OSSEs and real case studies often use an ensemble Kalman filter (EnKF; Evensen, 1994, 2003) to create the

forecast initial conditions (e.g., Snyder and Zhang, 2003; Dowell et al., 2004; Snook et al., 2011; Romine et al., 2013; Wheatley et al., 2015; Jones et al., 2016; Johnson and Wang, 2017). This strategy is preferred for CAM forecast experiments because the data assimilation system can update unobserved model state variables using flow-dependent error covariances derived from the forecast ensemble. The EnKF can also assimilate remotely sensed observations (e.g., radar reflectivity) and use cross-covariances to update environmental and in-storm fields (e.g., Snyder and Zhang, 2003). To maximize data assimilation system





performance, experiments must craft a forecast ensemble that is representative of the event uncertainty. This is challenging when the initial state is generated from a single or composite sounding. Therefore, a variety of strategies have been used to create the initial forecast ensemble. For example, random perturbations are commonly used to add variability to the initial environment to create the ensemble. The perturbations are inserted into the environment as gridpoint noise (e.g., Snyder and Zhang, 2003; Tong and Xue, 2005; Dawson et al., 2012) or smooth spatially-correlated structures (e.g., Caya et al., 2005;

Dowell and Wicker, 2009; Jung et al., 2012) to create the ensemble. The perturbation amplitude is calibrated to represent sources of uncertainty, including environmental variability (e.g., Dawson et al., 2012) and model error (e.g., Cintineo and Stensrud, 2013). While these perturbations are calibrated to account for sources of ensemble uncertainty, they remain an ad hoc technique used to introduce uncertainty into the forecast. The evolution of ensemble spread is sensitive to many user-defined parameters including perturbation length scale, amplitude, and location (e.g., Snyder and Zhang, 2003; Dowell et al.,

2004; Caya et al., 2005).

Despite being commonly used, random initial condition perturbations are not necessarily representative of the forecast errors observed in real case studies. Many studies assume the perturbations are unbiased and have little impact on the mean ensemble environment. This contradicts many previous CAM studies that note model errors bias the forecast environment (e.g., Snook and Xue, 2008; Coniglio et al., 2013; Romine et al., 2013; Cohen et al., 2015). Since the ensemble mean environment closely

matches the observed truth, OSSEs are unable to determine what impact assimilated environmental observations (e.g., surface stations, aircraft soundings) could have on the storm forecasts. To confront this challenge, some studies strategically bias the forecast initial environment (e.g., Stratman et al., 2018) or purposely fail to initiate convection (e.g., Kerr et al., 2015) to ensure greater disparity between the nature run simulation and forecasts. Although these studies better represent forecast biases observed in real case studies, there remain few options to introduce appropriate forecast biases in an OSSE.

A diverse set of OSSE cases that challenge the effectiveness of our data assimilation systems is required to holistically understand the impact observing systems have on CAM forecasts. The goal of this study is to provide a novel OSSE framework to understand how assimilated observations impact forecast performance. This study provides instructions to create an OSSE for a QLCS that initiates along a frontal boundary in a high-shear low-instability environment. Techniques to create the initial ensemble of simulations, which rely on uncertainties introduced by model physics, are also introduced. The steps taken to

create the nature run and forecast ensemble are listed in sections 2 and 3. Sections 4 and 5 describe the data assimilation procedure and forecast verification metrics. An example OSSE is conducted in section 6, and section 7 discusses the use of this OSSE framework for future studies.

## 2 Nature Run Configuration

### 2.1 Initial Environment

The nature run is initialized with a horizontally homogenous environment using a high-shear, low-convective-available-potential-energy (CAPE) composite sounding. The initial sounding (Fig. 1a), introduced by Sherburn and Parker (2019), is modified in the lower troposphere to support the development of robust boundary layer turbulence. High-shear, low-CAPE environments,



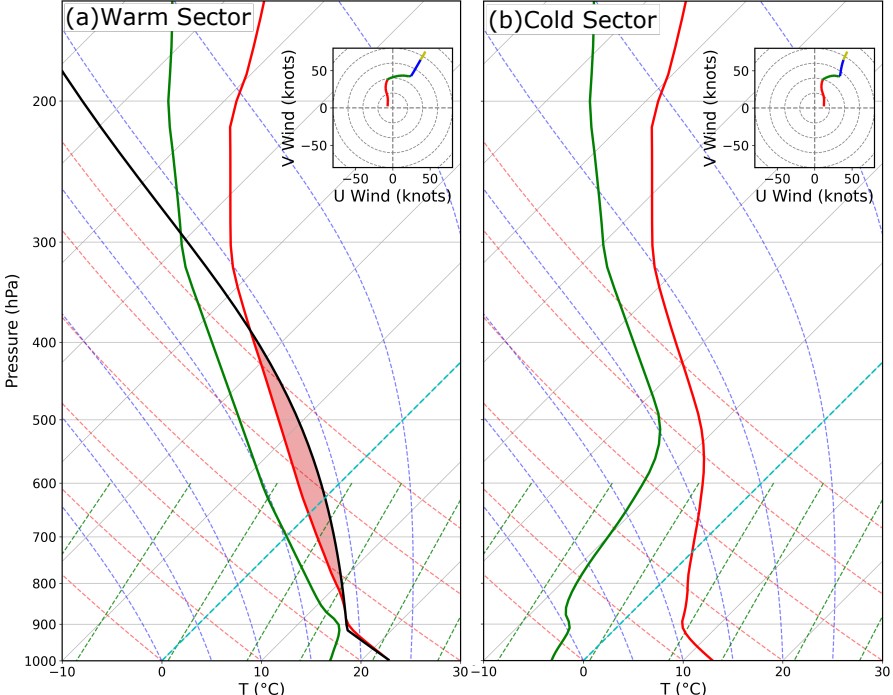

**Figure 1.** (a) The initial sounding for the nature run simulation. Red and green vertical lines correspond with T and $T_d$. The black vertical line marks the temperature of an air parcel launched from the surface. The hodograph, which plotted in the upper-right corner, is color coded by height above ground level (AGL): $0 - 1$ km is red, $1 - 3$ km is green, $3 - 5$ km is blue, $5 - 10$ km is yellow. Both soundings also initialize the (a) warm sector and (b) cold sector simulations that create the forecast ensemble.

which support approximately half of the Contiguous United States (CONUS) significant tornadoes (EF2+) (Schneider et al.,
2006), are the primary target of Verifications of the Origins of Rotation in Tornadoes Experiment-Southeast (VORTEX-SE)
field experiments. OSSEs initialized with this environment should help determine which observing systems benefit forecast
skill most and how to appropriately deploy them (e.g., spatial and temporal density).

A cold front provides the mechanical forcing required to initiate sustained convection for this case. Potential temperature
($\theta$), dewpoint temperature ($T_d$), and *u*-wind perturbations of -10 K, -20 K, and 10 m s$^{-1}$, respectively, are inserted along the
western domain edge to create the frontal boundary. Perturbation magnitude $f(x)'$ decreases with distance from the western and
bottom domain boundary following a cosine function:

$$f(x)' = perturbation \times cos(\frac{\pi\beta}{2})^2 \tag{1}$$

$\beta$ is defined as:

$$\beta = \sqrt{(\frac{x}{front\,width + f(y)})^2 + (\frac{z}{front\,height})^2} \tag{2}$$





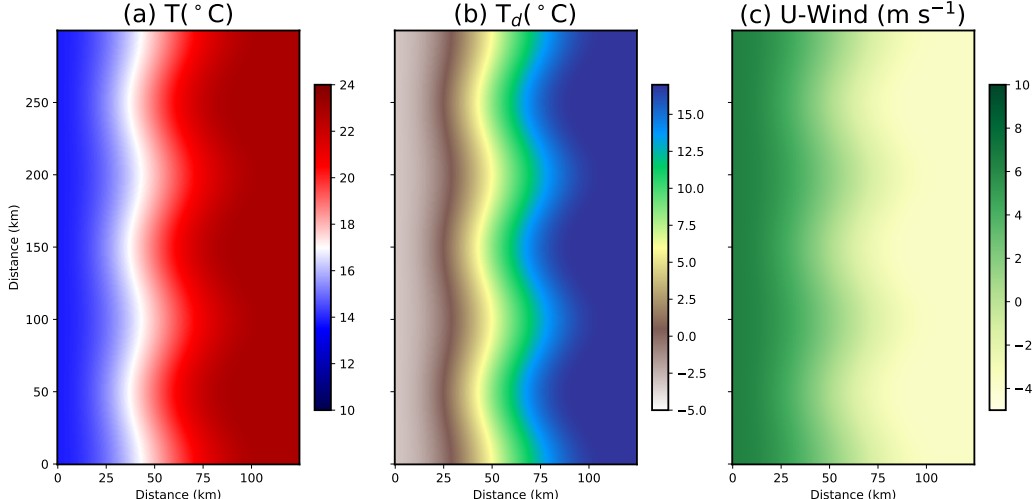

**Figure 2.** Nature run initial conditions at the lowest model level for (a) T, (b) $T_d$, and (c) *u*-wind. The plotted subdomain is centered upon the initial frontal boundary.

where x, y, and z are the distance of a grid point from the western, southern, and bottom boundaries, respectively. The assumed
frontal width is 100 km and height is 6 km. These equations generate a north-south front, that left undisturbed, will cause storms to initiate at the same time and east-west location. To vary convective initiation timing and location, waves are added to the frontal boundary in the north-south direction via the function *f(y)*:

$$f(y) = \delta cos(\frac{2\pi y}{\lambda} - \phi) \tag{3}$$

where $\lambda$ is wavelength (100 km), $\phi$ is phase shift (0), and $\delta$ is wave amplitude (10 km). The subtle frontal waves (Fig. 2)
simulate the natural variability observed in frontal location.

## 2.2 Prediction Model Settings

The nature run simulation for this case is created using the Cloud Model One (CM1; Bryan and Fritsch, 2002; Bryan and Rotunno, 2009) release 20.1. The simulation is run between 0000 – 0800 UTC on 1 January 2021 on a domain that centered upon Jackson, Mississippi (32.30 ºN, -90.18 ºW). The nature run time and location correspond with previous cold season
tornadic events that occur in the Southeastern United States (Sherburn and Parker, 2014). The experiment is run at 500 m horizontal grid spacing over a domain that spans 600 x 1200 computational points in the horizontal. The horizontal grid spacing, which is too coarse to fully resolve large turbulent eddies (Bryan et al. 2003), was selected to keep the relatively large domain computationally feasible and still partially resolve boundary layer turbulence. Lateral boundary conditions are open in the x-direction to preserve the east-west temperature gradient and periodic in the y-direction. No Coriolis acceleration is
assumed due to the complex nature of the atmospheric flow associated with the fontal boundary.



The 120-level vertical grid is stretched from 10 m at the lowest model level to 250 m at heights between 9.875 – 15 km (i.e., the model top). The simulation is run with a semi-slip bottom boundary condition and uses the Jiménez et al. (2012) surface layer scheme to calculate surface fluxes and surface stress. The upper boundary condition is free slip; Rayleigh damping with a coefficient of 0.003 s$^{-1}$ is applied starting at 12 km above ground level (AGL).

Model physics options for this case were selected to ensure the simulation accurately portrays the evolution of a storm system. Precipitation processes are parameterized using the double-moment Morrison microphysical parameterization (Morrison et al., 2005, 2009). The microphysical parameterization predicts the evolution of a single rimed ice species similar to hail (e.g., dense, faster fall speeds), which produces realistic squall line simulations (Bryan and Morrison 2012). The NASA-Goddard radiative scheme simulates the effects of longwave and shortwave radiative forcing during the simulation period. Subgrid-scale
turbulence is parameterized using the Deardorff (1980) turbulent kinetic energy scheme.

## 2.3    Boundary Layer Turbulence

Boundary layer turbulence plays an important role in the evolution of convection. Storm interactions with turbulent eddies can modify the mesocyclone circulation as well as storm intensity and location (e.g., Nowotarski et al., 2015; Markowski, 2020; Labriola and Wicker, 2022). Turbulent eddies transport near-surface air further aloft and impact boundary layer temperature,
wind, and moisture profiles. The turbulence also facilitates the downward transport of high momentum air aloft and is consequently necessary to properly simulate the effects of surface friction on thunderstorm evolution (Markowski, 2016). To enhance experiment realism, the nature run simulation for this case is initialized with fully mature boundary layer turbulence using a technique introduced by Markowski (2020) and subsequently improved upon by Labriola and Wicker (2022).

     To introduce realistic eddies, a second simulation, referred to as the "turbulence simulation", is used to create a realization
of boundary layer turbulence. The turbulence simulation is initialized with the nature run initial sounding (Fig. 1a) on the same grid configuration. Small-scale pseudo random $\theta$ perturbations ($\pm$ 0.25K) are superimposed on the environment. As the turbulence simulation is integrated 12 hours forward in time, the random perturbations evolve to form a turbulent boundary layer (Fig. 3). Once the turbulence simulation is complete, the perturbation $u$, $v$, $w$, $\theta$, and water vapor mixing ratio ($q_v$) fields are superimposed on the nature run initial conditions. Perturbation fields are the difference between a model state variable and
the horizontal plane domain average. Since the turbulent perturbations average to zero across the domain, they have no impact on the nature run mean initial environment.

     Turbulence simulation settings are nearly the same as the nature run configuration, with a few notable exceptions. The lateral boundary conditions are periodic in all directions so that robust turbulent eddies persist across domain boundaries. The simulations used to generate turbulence also assume no radiative forcing, no surface friction, and apply Coriolis acceleration to
the perturbation wind field. These settings are necessary to form robust turbulent motions (Fig. 3) without spawning spurious convection or substantially modifying the initial environment. See Labriola and Wicker (2022) for more details.



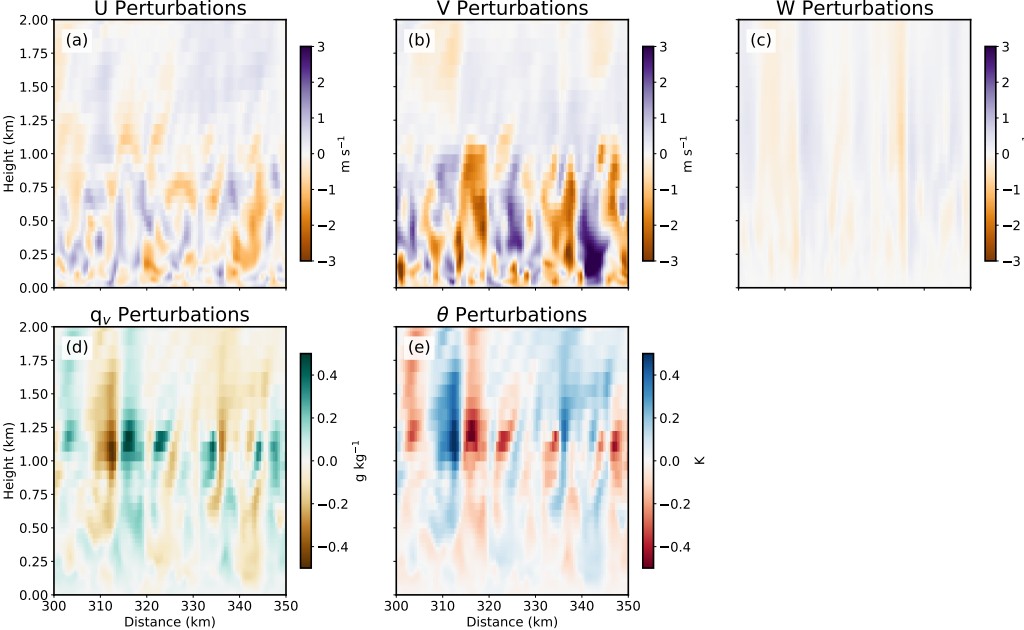

**Figure 3.** Vertical cross-sections of boundary layer turbulent perturbations that are inserted into the nature run simulation. Plotted fields include the (a) $u$- (b) $v$-, and (c) $w$-components of wind, as well as (d) $q_v$, and (e) $\theta$.

## 2.4 Simulation Results

The nature run setup produces a QLCS that persists for several hours before exiting the experiment domain (Fig. 4). Although the cold air perturbation alone can initiate robust convection (e.g., Sherburn and Parker, 2019; Labriola and Wicker, 2022), the positive $u$-wind perturbation along the western domain boundary is necessary to sustain convection. The post-frontal winds advect cold air eastward with time (Fig. 4a, c, e) and maintain strong temperature and moisture gradients along the boundary. The wind perturbation also enhances convergence, which initiates and sustains the robust storm system.

QLCS storm structure changes during the simulation period. During the first two hours of the simulation (e.g., Fig. 4b) the convective line is robust and storms are spaced closely together so that it is difficult to discern individual convective cores. As the simulation progresses, the QLCS embedded storms become more isolated and the trailing region of stratiform precipitation expands in areal coverage (Fig. 4d, f). Boundary layer turbulence and QLCS modifications to the environment also cause isolated storms to initiate in the warm sector (Fig. 4f). Strong vertical wind shear and modest instability cause many of the isolated storms to persist until they exit the domain or are absorbed by the approaching QLCS.

Severe weather hazards (e.g., tornadoes, wind, hail) are small in scale and not fully resolved by the nature run simulation. Since these phenomena cannot be explicitly predicted, diagnostic tools are used to identify areas of intense convection in model output. Updraft helicity, which is the vertical integration of updraft intensity multiplied by vertical vorticity between 2 and 5 km (Kain et al., 2008), is a commonly used proxy for severe weather in CAM forecast experiments (e.g., Kain et al., 2010;



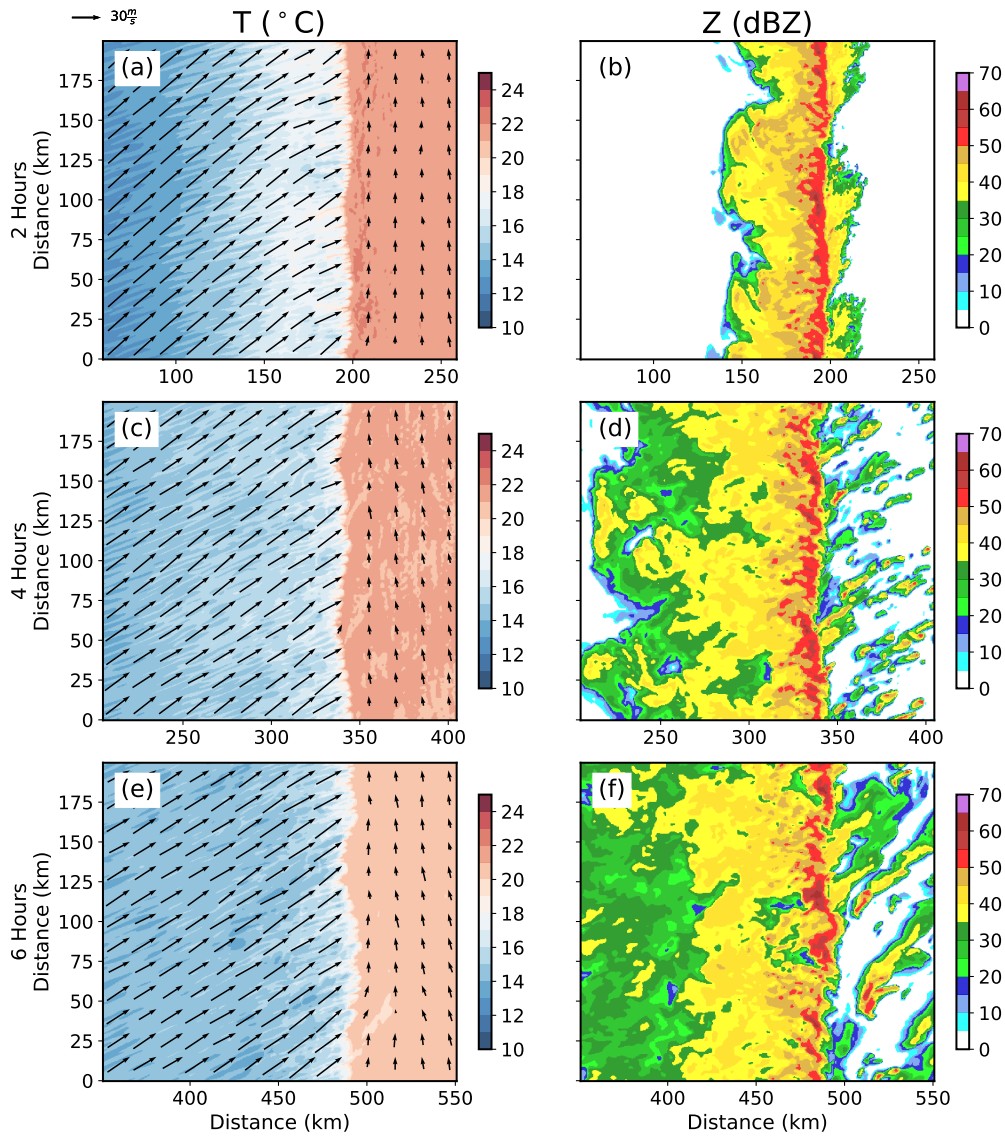

**Figure 4.** Nature run (a,b,c) T and (d,e,f) Z at the lowest model level (5 m AGL). 500 m AGL winds are superimposed on (a,c,e) using arrows.

Sobash et al., 2011, 2016; Clark et al., 2012; Gallo et al., 2016; Loken et al., 2017; Carlin et al., 2017; Skinner et al., 2018; Potvin et al., 2019; Miller et al., 2022). Updraft helicity can skillfully predict severe weather events because the algorithm
identifies mid-level mesocyclones that produce a disproportionately large number of severe weather reports. To identify areas of intense convection and understand how these storms evolve, the nature run maximum updraft helicity is evaluated.




The nature run produces several long-track swaths of enhanced updraft helicity ($> 500$ m$^2$ s$^{-2}$), which suggests some QLCS embedded storms are capable of producing severe weather. QLCS updraft helicity swaths (Fig. 5a) are short-lived and numerous in the western half of the experiment domain, where the storm system initiates and individual storm cells frequently

interact (Fig. 4b). Later in the simulation, as the QLCS convective cores become more diffuse (Fig. 4d), the number of updraft helicity swaths decreases (Fig. 5a; 300 km < x < 450 km). In the eastern quarter of the experiment domain (x > 400 km), isolated storms initiate ahead of the QLCS and produce long uninterrupted updraft helicity swaths (Fig. 5c). The rotating storms initially move northeasterly until interacting with the QLCS; after which the updraft helicity intensity decreases and the swaths rotate eastward. The complex storm interactions for this case highlight the challenge to predict severe weather hazards

associated with an evolving QLCS.

## 3    Forecast Ensemble Configuration

Rather than using random perturbations to generate the initial ensemble, this study relies upon uncertainties introduced by model physics. The following steps are taken to create each of the 40 ensemble members for this case:

– Step 1: Initialize two separate horizontally homogenous simulations with a warm sector environment (Fig. 1a) and a cold
sector environment (Fig. 1b)

– Step 2: Select a model surface type and insert random ($\pm$ 0.25 K) $\theta$ perturbations

– Step 3: Run both simulations for 24 hours

– Step 4: Blend the simulations together to recreate the initial frontal boundary

The remainder of this section explains, in depth, the procedure used to generate forecast initial conditions.

### 3.1    Ensemble Initialization

Simulations are run with different land surfaces to allow the atmosphere to evolve freely and generate the 40-member forecast ensemble. To avoid spawning the QLCS early, two separate simulations are created for each ensemble member to predict the evolution of the air mass ahead (i.e., warm sector) and behind (i.e., cold sector) the front. The warm sector simulation is initialized with the nature run initial sounding (Fig. 1a). $\theta$, T$_d$, and $u$-wind perturbations consistent with the nature run (-10 K,
-20 K, 10 m s$^{-1}$, respectively) are added to the initial sounding to create the cold sector environment (Fig. 1b). Perturbation amplitude decreases as a cosine function of height above the surface and extends 6 km AGL. Once both simulations are initialized, pseudo-random potential temperature perturbations ($\pm$ 0.25 K) are inserted into the environments to further encourage ensemble diversity.

Parameterized air-surface interactions are a substantial source of forecast uncertainty. Land surface conditions impact the
boundary layer and can subsequently alter storm evolution (e.g., Reames and Stensrud, 2017, 2018; Yang et al., 2021). Consequently, the heterogeneous surface makeup of the Southeastern United States can modify the environment in countless ways. To incorporate these uncertainties into the ensemble design, cold and warm sector simulations for each ensemble member are

 

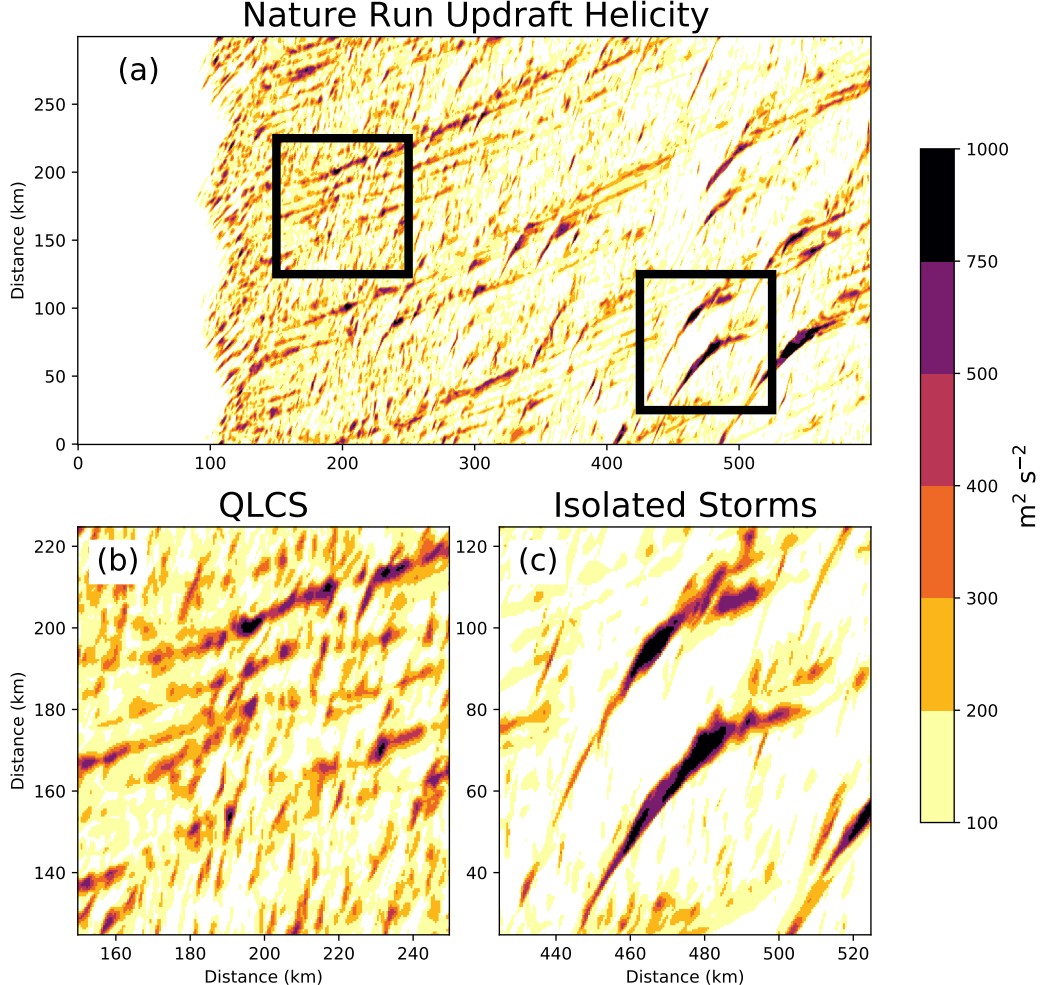

**Figure 5.** Nature run maximum updraft helicity over the (a) full experiment domain. Subdomains plotted in (b) and (c) highlight QLCS embedded mesovortices (left black box) and isolated convection (right black box), respectively.

assigned a land surface type that is commonly observed in the Southeast (Table 1). Simulated land surfaces include: various degrees of suburban and urban sprawl, croplands, grasslands, forests, bogs, and open bodies water (assumed in warm sector simulations only). Both simulations are then integrated for 24 hours so that surface dependent momentum, heat, and moisture fluxes can modify the lower troposphere.

Once cold and warm sector simulations are complete, they are blended together to form the initial cold front boundary that initiates the QLCS. A cosine weighting function consistent with eqn. 1 blends the air masses. The cold sector simulation solution is given full weight along the western domain boundary. The warm sector solution is increasingly favored eastward and given full weighting at all locations east of 100 km. To initiate convection at different times and locations, small changes



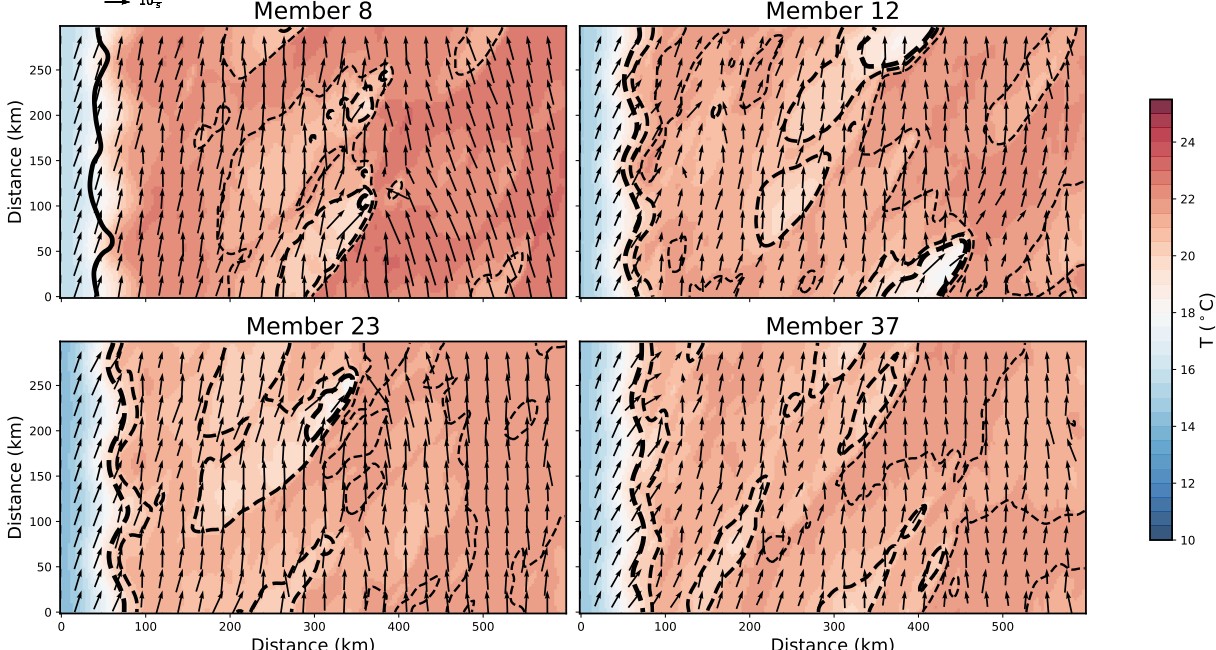

**Figure 6.** Forecast air temperature at the lowest model level (25 m AGL) at the time of ensemble initialization (0000 UTC). Forecast wind speed sampled at 500 m AGL are plotted with arrows. The difference between the forecast and nature run temperature are contoured in 1 °C increments. Contours that are dashed (solid) indicate where the forecast is cooler (warmer) than the nature run. Contours become thicker as error increases.

are made to the frontal boundary width (100 km $\pm$ 5 km), the number of frontal waves (3 $\pm$ 0.5), as well as wave amplitude (10 km $\pm$ 5 km) and phase ($\pm \frac{\pi}{4}$). The initial conditions for each ensemble member (Fig. 6) resembles the nature run; however, given the 24-hour spin-up there is greater uncertainty in environmental conditions.

## 3.2 Prediction Model Settings

The forecast ensemble is designed to resemble the Warn-on-Forecast System (WoFS; Wheatley et al., 2015; Jones et al., 2016), a frequent-updating CAM forecast ensemble that predicts the evolution of severe weather events in real-time that is run by the National Severe Storms Laboratory. This strategic choice allows OSSEs to understand how assimilated observations impact a real-time ensemble prediction system that provides useful guidance to the operational community (e.g., Wilson et al., 2021; Gallo et al., 2022). Forecasts are run on a 100 x 200-point grid with 3 km horizontal grid spacing and 50 vertical levels. Vertical
grid spacing is stretched, with the smallest grid spacing ($\Delta z$ = 50 m) located at the surface and the coarsest grid spacing ($\Delta z$ = 550 m) located at the 15 km model top. Forecast prediction model settings are the same as the nature run, except the Yonsei University (YSU) scheme (Hong et al., 2006) parameterizes the effects of boundary layer turbulence. Forecasts are run using release 18 of the CM1 prediction system. Although the forecast ensemble and nature run are both generated using the CM1,



**Table 1.** The surface types used to generate the initial ensemble of forecasts. Lu0 is the initial land-use index in CM1 namelist.

| Members | Lu0 | Members | Lu0 |
|---------|-----|---------|-----|
| 1, 21   | 1   | 11, 31  | 12  |
| 2, 22   | 2   | 12, 32  | 13  |
| 3, 23   | 3   | 13, 33  | 14  |
| 4, 24   | 4   | 14, 34  | 15  |
| 5, 25   | 5   | 15, 35  | 16 (Cold Sector = 33) |
| 6, 26   | 6   | 16, 36  | 17  |
| 7, 27   | 7   | 17, 37  | 18  |
| 8, 28   | 8   | 18, 38  | 19  |
| 9, 29   | 9   | 19, 39  | 31  |
| 10, 30  | 10  | 20, 40  | 32  |

they are produced using different releases. Differences between prediction model releases and other factors (i.e., model physics
configuration and grid spacing) are expected to mitigate the "identical twin" problem that can cause OSSE results to become
overly optimistic (e.g., Hoffman and Atlas, 2016).

### 3.3   Forecast Results

The initial forecast ensemble represents some of the complexities observed in real-data cases. Although most data assimilation
systems make use of Gaussian prior error approximations, the non-linear error growth attributed to mesoscale processes often
cause forecast errors to become non-Gaussian. Data assimilation systems can produce posterior state estimates under these
conditions, but subsequent analyses and forecasts are often suboptimal (e.g., Poterjoy et al., 2017; Robert et al., 2018; Buehner
and Jacques, 2020; Poterjoy, 2022). Due to the evolution of mesoscale processes in the warm and cold sector environments,
forecast errors for this OSSE are non-Gaussian (e.g., Fig. 7) and more realistically challenge the performance of the data
assimilation system.
Errors introduced by the model physics uncertainties cause the ensemble to drift away from the nature run environment.
During the 24-hour simulations, storm cold pools and increased cloud cover cause the warm sector surface layer to become
moister and cooler on average (Fig. 7a). Relatively warm ground temperatures moisten and heat the lower troposphere in the
cold sector (Fig. 7b). Friction also modifies the environment and slows near-surface winds (Fig. 7). Due the idealized nature
of these simulations, there is no pressure gradient force to counteract the impacts of friction, so surface winds are nudged
to zero. Ensemble wind spread is larger in the warm sector simulation (Fig. 7a) in part because convective storms disrupt the
environment. Moderating cold and warm sector environments cause the atmosphere to become more stable and weakens frontal
intensity (i.e., weaker gradients, less convergence), which impacts the strength of the subsequent forecast QLCS.







**Figure 7.** Profiles of the domain average (a) warm sector and (b) cold sector environments used to generate the initial ensemble. Red, green, and black lines correspond with T (°C), T$_d$ (°C), and wind (knots). 40 ensemble member profiles are marked with a thin translucent line. The profiles used to initialize warm sector and cold sector simulations (i.e., the unperturbed soundings) are marked by a bold line.



**Figure 8.** (a) Nature run and (b-d) forecast $Z$ at the lowest model level at 0300 UTC. The forecasts are integrated forward in time without data assimilation.

Modifications to the environment cause forecast storms to be weaker and less numerous than the nature run simulation (Fig. 8). Some members predict discrete convective storms to form near the observed QLCS (Fig. 8b, d), but fail to form an organized line of storms. This occurs because moderating warm and cold sector environments weaken the frontal boundary temperature gradient. Further, weaker winds dimmish convergence that initiates the line of storms. It is noted that some ensemble members predict a QLCS (Fig. 8c), but the storm system is smaller and weaker than the nature run (Fig. 8a). Differences are attributed to changes in the horizontal grid spacing, which impacts storm updraft intensity and areal coverage (e.g., Bryan et al., 2003; Bryan





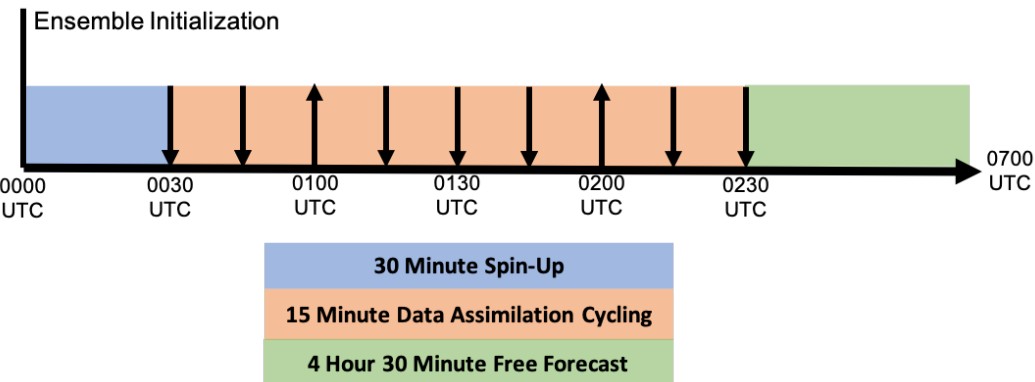

**Figure 9.** The OSSE data assimilation timeline. Downward pointing arrows indicate times when only radar observations are assimilated. Upward arrows indicate when surface and radar observations are assimilated.

and Morrison, 2012; Verrelle et al., 2015). Additionally, the environment is more stable and has weaker low-level convergence, so the forecast QLCS is less intense than the nature run.

## 4 Data Assimilation Procedure

The data assimilation configuration for this study is designed to resemble the WoFS (Fig. 9). After initialization, the 40-member ensemble of forecasts undergoes a 30-minute spin-up period until 0030 UTC when observations are assimilated using the Data Assimilation Research Testbed (DART; Anderson and Collins 2007; Anderson et al. 2009) ensemble adjustment Kalman filter (EAKF; Anderson 2001). Observations are assimilated every 15 minutes over a 2-hour window between 0030 – 0230 UTC. Following data assimilation, the forecast ensemble is run until 0700 UTC (4.5-hour forecast) before the QLCS exits the experiment domain.

Simulated radar, surface, and sounding observations extracted from the nature run simulation are assimilated during this study. Four radar sites that are spaced approximately 240 km apart in the experiment domain (Fig. 10) provide simulated reflectivity ($Z$) and radial velocity ($V_r$) observations. The radar observations are interpolated in the vertical direction to generate 14 tilts that are consistent with the next generation weather radar (NEXRAD; Crum et al., 1993) system scanning pattern. A Cressman weighting function (Cressman, 1959) with a 3 km radius of influence analyzes observations to a 5 km grid in the horizontal direction. Observations within 150 km of the parent radar site are assimilated. Radar observations update all model state variables except surface moisture and skin temperature fields.

Radar data assimilation alone does necessarily result in optimal forecast performance due to more realistic and non-Gaussian initial condition errors. This is different from many previous CAM OSSEs that produce skilled forecasts after assimilating only radar observations (e.g., Snyder and Zhang, 2003; Dowell et al., 2004; Caya et al., 2005; Jung et al., 2008; Potvin et al., 2013;



**Table 2.** The data assimilation parameters used for each observation type.

|  | Radar | Surface | Sounding |
|---|---|---|---|
| Number of Sites | 4 | 54 | 4 |
| Assimilation Frequency | 15 Minutes | Top of Hour | Top of Hour |
| Observation Errors | 6 dBZ (Z) $3$ m s$^{-1}$ ($V_r$) | 1.75 m s$^{-1}$ ($u,v$) 1.5 K (T) 2.0 K (T$_d$) | 1.75 m s$^{-1}$ ($u,v$) 1.5 K (T) 2.0 K (T$_d$) |
| Localization Radius | 12 km (Horizontal) 6 km (Vertical) | 150 km (Horizontal) 4 km (Vertical) | 500 km (Horizontal) 4 km (Vertical) |

Stratman et al., 2018). Conventional observations must be also assimilated in this study to improve forecast skill. This is consistent with operational forecast systems where multiple observation sources are always assimilated.

Simulated soundings (i.e., an instantaneous profile of the atmosphere) are assimilated at the top of each hour from each radar site (average site spacing = 243.5 km). Soundings sample the atmosphere every 100 m and provide observations of air temperature (T), dewpoint temperature (T$_d$), and $u$- and $v$-wind. Simulated surface observations mimic what is reported by the automated surface observing system (ASOS) network in real-time. Surface observations are assimilated at the top of the hour (Fig. 9) and include 2 m temperature, 2 m dewpoint temperature, and 10 m $u$- and $v$-wind. 54 surface stations are randomly

distributed throughout the experiment domain to match the approximate spatial density of ASOS stations in the southeastern United States (Fig. 10). Errors are added to each assimilated observation (radar, soundings, and surface) by randomly drawing perturbations from a zero-mean gaussian distribution that is equivalent to the observation error variance (Table 2).

Data assimilation systems are subject to sampling errors that can cause observations to become spuriously correlated with distant model state variables. Left unchecked, the assimilated observations will degrade the analyzed model state and limit

forecast skill. Covariance localization mitigates this problem by limiting the radius over which an assimilated can impact the model state. This study uses a distance-based Gaussian weighting function (Gaspari and Cohn, 1999) to limit the range of influence. Localization radii for the assimilated observations closely resemble what is employed by the WoFS (Table 2). Spatially and temporally varying adaptive inflation (Anderson and Collins, 2007) is applied to the prior ensemble to maintain ensemble spread during data assimilation. Inflation parameters are defined in Table 3.

**5   Forecast Verification**

Observation space statistics provide insight into data assimilation system performance. The root-mean square innovation (RMSI), which is the difference between the ensemble mean prior or posterior and the nature run simulation, quantifies the fit of forecasts and analyses to the observed environment. Rather than calculating statistics at the location of an observing station,





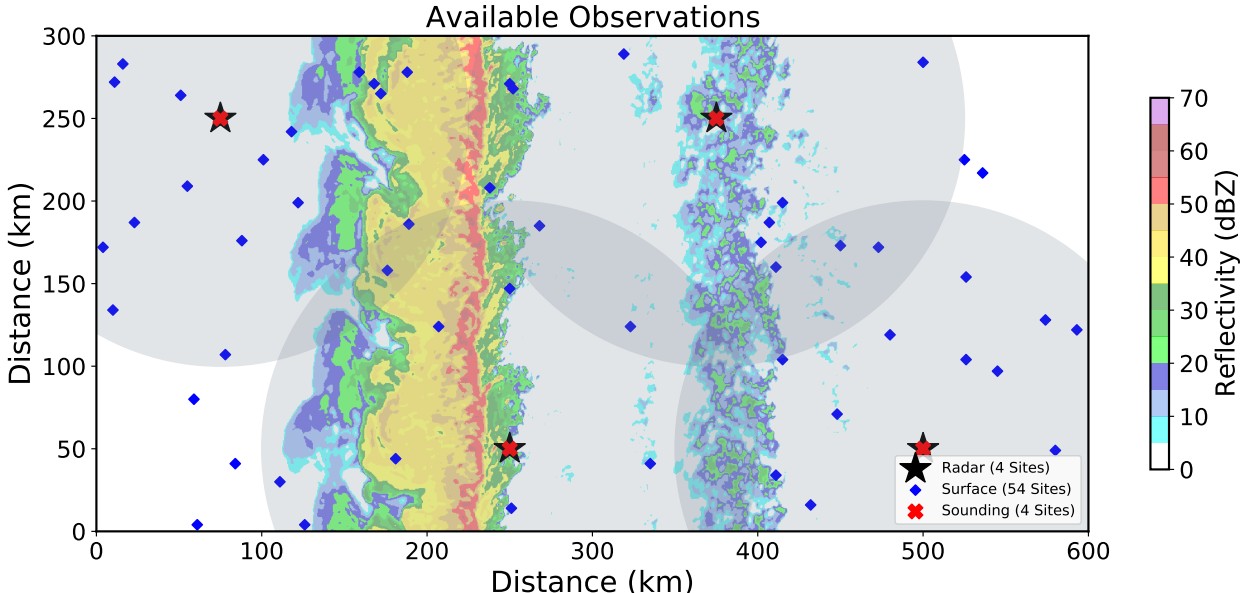

**Figure 10.** The experiment domain with assimilated observations plotted. Gray shaded circles mark scanning radius of four radars (marked by black stars). Red crosses mark the sites where soundings are launched. 54 assimilated surface stations are marked by blue diamonds. Nature run $Z$ at the time of the final data assimilation cycle (0230 UTC) is plotted for reference.

**Table 3.** The Surface types used to generate the initial ensemble of forecasts. Lu0 is the initial land-use index in CM1 namelist.

| Inflation Parameter | Defined Value |
|---|---|
| inf_initial | 1.0 |
| inf_sd_initial | 0.6 |
| inf_damping | 0.9 |
| inf_lower_bound | 1.0 |
| inf_upper_bound | 100 |
| inf_sd_lower_bound | 0.6 |

innovations are calculated for environmental fields (T, $T_d$, $u$, $v$) over the lowest 3 km of the forecast domain. This region of the atmosphere is where forecast errors are largest, and provides insight into how assimilated observations impact the environment.

Forecast and nature run simulated radar $Z$ sampled at the lowest model level are compared to evaluate QLCS intensity, position, and structure. The neighborhood maximum ensemble probability (NMEP Schwartz and Sobash, 2017) of forecast $Z$ exceeding 45 dBZ (P[$Z$ > 45 dBZ]) evaluates storm cores embedded within the QLCS. To mitigate small displacement errors, a 9 km neighborhood is used to generate the probabilistic fields. A Gaussian filter with the same radius smooths the subsequent






forecast probabilities. Probabilistic forecast guidance is subjectively compared and objectively verified against the nature run to measure forecast skill.

The Brier skill score (BSS; Brier, 1950) objectively quantifies probabilistic forecast skill for this study. This measurement of skill, which ranges between values less than zero (no skill) and one (perfect skill), can be decomposed into reliability, resolution, and uncertainty (Murphy, 1973):

$$BSS = \frac{Resolution - Reliability}{Uncertainty} \tag{4}$$

Reliability is the difference between forecast probability and the relative frequency of the event occurring for that given probability threshold. Forecast skill is optimized when this difference is minimized. Resolution is the difference between the observed climatology of an event occurrence and the frequency that a forecast event occurs for a given probability threshold. Forecast skill increases with resolution. While the first two parameters are defined by forecast performance, uncertainty is a measure of event climatology. Attributes diagrams, which plot forecast probability against observed frequency, provide insight into forecast reliability and resolution.

## 6   Forecast and Data Assimilation Experiment

Two forecast and data assimilation experiments are run to demonstrate of impact assimilated environmental and radar observations. The control experiment (CTRL) assimilates only radar observations. The second experiment (ENVI) assimilates radar, surface, and sounding observations.

CTRL RMSI values (Fig. 11) are constant or increase after successive data assimilation cycles. Although cross-covariances allow assimilated radar observations to update the model state, the impact of radar observations is confined to regions near or within convection. Much of the domain is clear air during the data assimilation window (Fig. 10), so radar observations have a limited impact on the broad environment. Assimilated environmental observations substantially reduce wind field errors (Fig. 11a-b), but have little impact or increase T (Fig. 11c) and $T_d$ (Fig. 11d) innovations. Assimilated sounding observations assume a large localization radius (Table 2) that is consistent with the Warn-on-Forecast System configuration. Large localization radii allow atmospheric observations to update large regions of the experiment domain; however, spurious correlations between distant model state variables can potentially increase error. Despite modest increases in error, RMSI values for temperature and moisture fields are comparable between both experiments and remain relatively small (i.e., less than or equal to the observation error variance) throughout data assimilation window.

Assimilated environmental observations enhance frontal boundary wind convergence and cause a more robust QLCS to form. The CTRL posterior $u$-wind field at the final assimilation cycle (Fig. 12b) is positively biased in the lowest 1 - 2 km of the warm sector (x > 250 km). This decreases CTRL frontal convergence because warm sector winds are directed eastward and away from the QLCS. Assimilated environmental observations decrease ENVI warm sector wind errors near the surface (Fig. 12c). In the cold sector (x < 225 km), ENVI winds are slightly stronger than CTRL, which decreases error (Fig. 12b-c). This further enhances convergence in ENVI forecasts and provides the mechanical forcing necessary to establish a QLCS that has larger updrafts (Fig. 13).



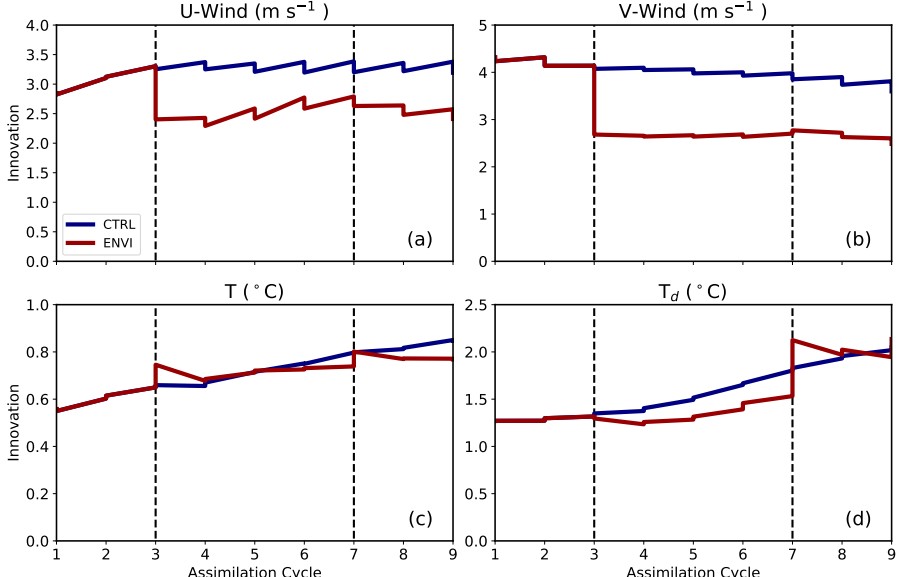

**Figure 11.** The domain averaged (a) $u$-wind (b) $v$-wind, (c) T, and (d) $T_d$ RMSI during the data assimilation window. Statistics consider the lowest 3 km of the troposphere where forecast errors are largest. Bold blue and red lines correspond with CTRL and ENVI, respectively. Vertical dashed lines mark the points when conventional observations (soundings and surface) are assimilated.

Experiment differences are more subtle for temperature (Fig. 12e-f) and moisture (Fig. 12h-i) fields. Both underpredict cold sector intensity; the near surface (< 0.5 km AGL) is too warm (Fig. 12e-f) and moist (Fig. 12h-i). ENVI cold sector biases

are smaller than CTRL, which suggests the assimilated environmental observations have a small but positive impact on the posterior state. Analyzed T and $T_d$ are similar in the warm sector (x > 250 km) for both experiments; ensembles are cool near the surface (Fig. 12 e-f) and dry further aloft (> 1 km AGL) (Fig. 12h-i). Despite localized errors, domain averaged temperature and moisture innovations are relatively small in magnitude (Fig. 11c, d).

Assimilated radar observations play an important role in the initial storm placement and intensity (e.g., Snyder and Zhang,

2003). Both experiments, which assimilate $Z$ and $V_r$ observations, predict high forecast probabilities (P[Z > 45 dBZ] > 0.7) to be collocated with observations early in the forecast (Fig. 14a, c). Consequently, both experiments have similar attributes curves (Fig. 14e).

CTRL forecast probabilities become displaced from observations over time (Fig. 14b) because the predicted QLCS moves too slowly. Forecast storm motion biases are commonly observed (e.g., VandenBerg et al., 2014) and are sensitive to many

factors including the wind profile errors and cold pool intensity. Due to storm displacement errors, CTRL forecast probabilities far exceed observed frequency (Fig. 14f) for high probability thresholds (P[$Z$ > 45 dBZ] > 0.6). The ENVI QLCS moves faster and is more closely located with the observed QLCS, causing the forecast ensemble to become more reliable (Fig. 14f). The areal coverage of high forecast probabilities (0.6 < P[$Z$>45 dBZ] < 0.8; Fig. 14f) is also higher for ENVI than CTRL. Forecast




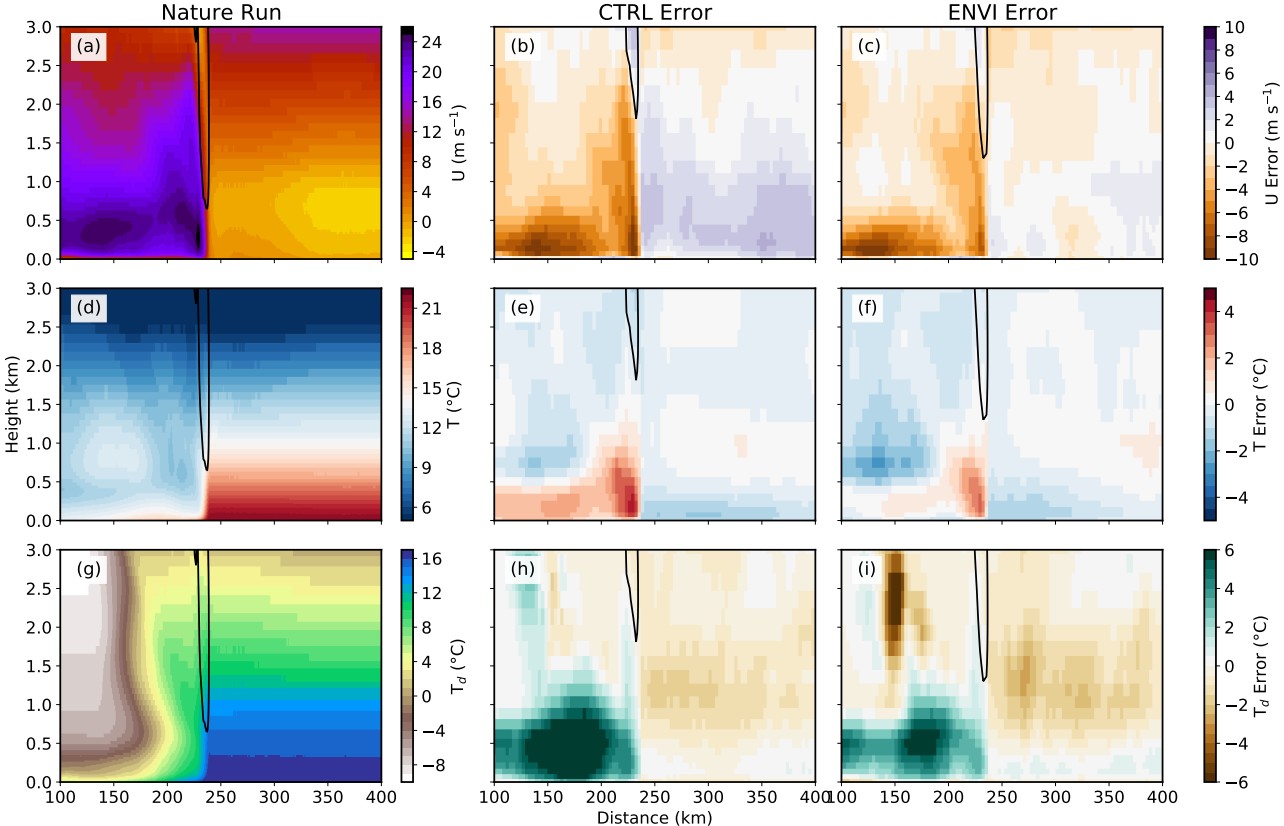

**Figure 12.** Ensemble mean posterior error at the time of the final data assimilation cycle (0230 UTC) for (a) *u*-wind, (b) T, and (c) T$_d$. Errors are averaged in the north-south direction. Regions where the north-south average updraft velocity exceeds 1.0 m s$^{-1}$ are contoured.

probabilities increase partly because ENVI forecasts predict storm updrafts to be larger than CTRL throughout much of the
forecast period (Fig. 13).

Forecast skill scores for both ensembles are similar early in the forecast period (Fig. 15), but diverge during the first two hours. The benefits of radar data assimilation wane during this time because small-scale errors quickly grow in scale and impact storm evolution (e.g., Aksoy et al., 2010). For example, CTRL skill decreases faster (Fig. 15) because the predicted QLCS moves too slow and becomes displaced from observations (Fig. 14b). The ENVI BSS is higher because the predicted
QLCS evolves in an environment that is more representative of the nature run (Fig. 11). Despite differences, both ensembles exhibit some skill (BSS > 0.0) during the first 4 hours of the forecast (Fig. 15). BSSs for both ensembles gradually decrease during this time because the QLCS becomes displaced from observations, the line of storms weakens, and storms ahead of the QLCS fail to initiate with sufficient coverage and intensity.



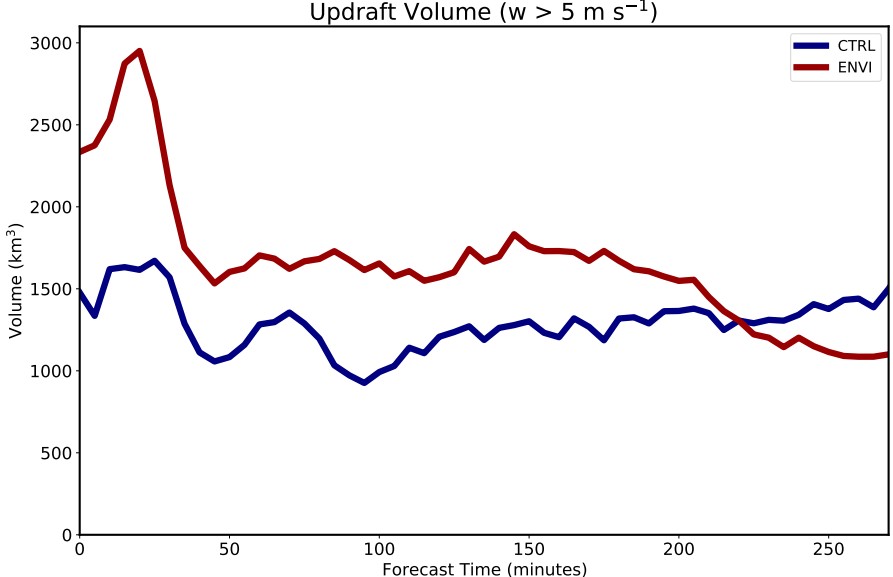

**Figure 13.** Ensemble mean updraft volume during the forecast period. Statistics only consider regions where updraft velocity exceeds 5 m s$^{-1}$. Bold blue and red lines correspond with CTRL and ENVI, respectively.

## 7 Summary

To gain a more robust understanding of how assimilated observations impact convection-allowing model (CAM) forecast skill, it is imperative that observing system simulation experiments (OSSEs) include a diverse range of case studies that simulate different storm modes and environments. Unfortunately, the number of idealized case studies for CAM OSSEs is limited. This paper introduces the techniques used to generate a nature run that is representative of a tornadic outbreak in the southeastern United States. The nature run simulates a cold front boundary that initiates and maintains a quasi-linear convective system

(QLCS) in a highly-sheared and modestly-unstable turbulent environment. During the 7-hour simulation, the QLCS produces multiple mesovortices and isolated rotating thunderstorms initiate ahead in the warm sector.

This study also introduces a new technique to create an ensemble of forecasts initialized from a single sounding. Uncertainties in the representation of surface conditions are leveraged to create the forecast ensemble. Each forecast member is initialized from the same sounding, but run for 24-hours assuming a different land surface type and small-scale random perturbations.

Different surface conditions alter surface heat, moisture, and momentum fluxes that modify the lower troposphere. The subsequent ensemble is representative of many real-data CAM case studies because the ensemble mean environment is degraded and the initial condition errors are non-Gaussian. This provides the opportunity to understand how assimilated environmental observations will impact forecast skill during a real case study.

The OSSE framework introduced in this study requires a combination of in-storm and environmental observations to create a

more skilled QLCS forecast. Forecasts that only assimilate radar observations (reflectivity, radial velocity) predict a relatively



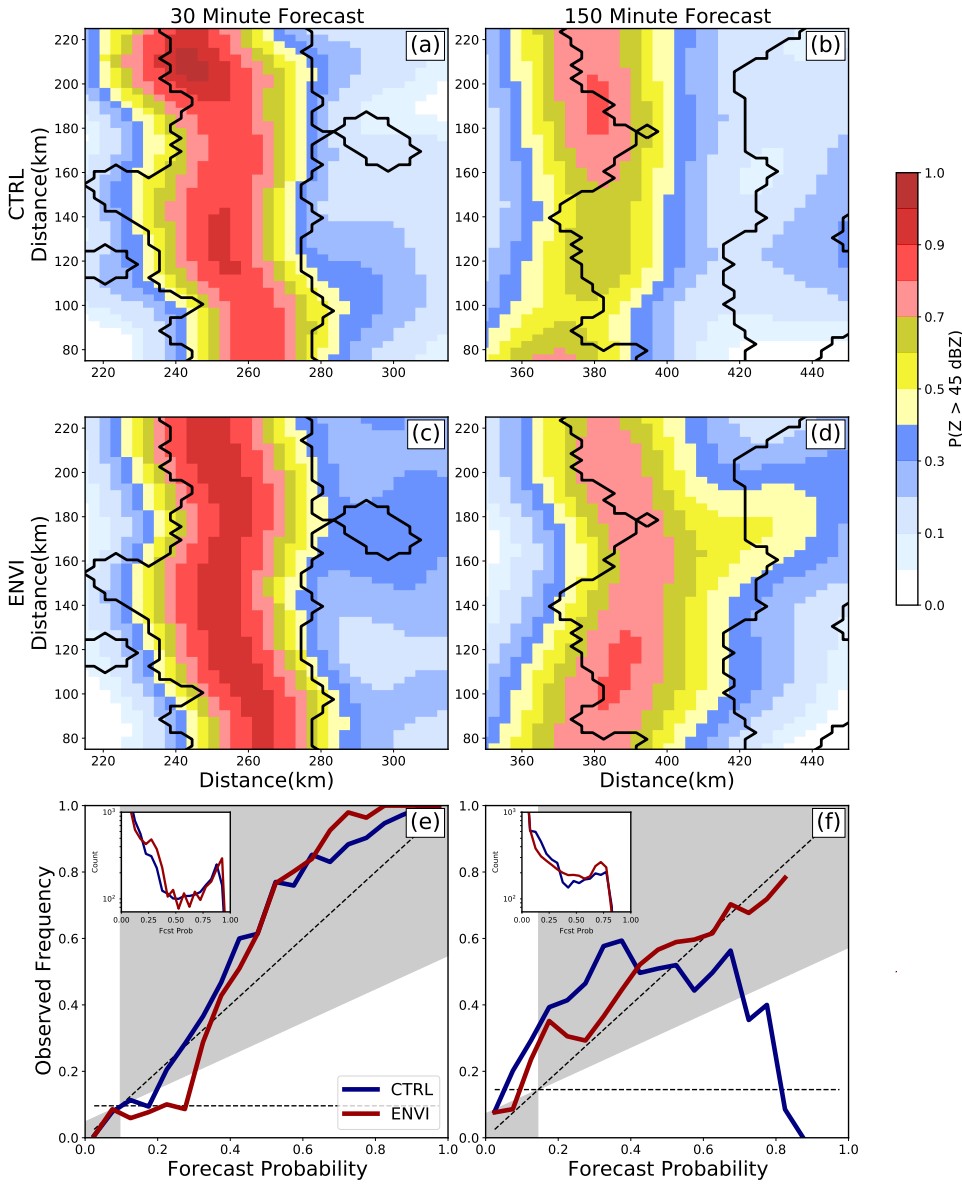

**Figure 14.** The P(Z > 45 dBZ) at (a, c) 30 and (b, d) 150 minutes for (a-b) CTRL and (c-d) ENVI. Regions where the nature run Z exceeds 45 dBZ are contoured black. Attributes diagrams evaluate forecast performance over the full experiment domain at (e) 30 and (f) 150 minutes.

weak QLCS that moves too slowly. This is in large part because surface friction weakens the initial ensemble wind profile in the lower troposphere which diminishes convergence along the front that initiates and sustains the QLCS. Environmental observations must also be assimilated to correct wind profile errors and increase convergence. Forecasts that assimilate radar



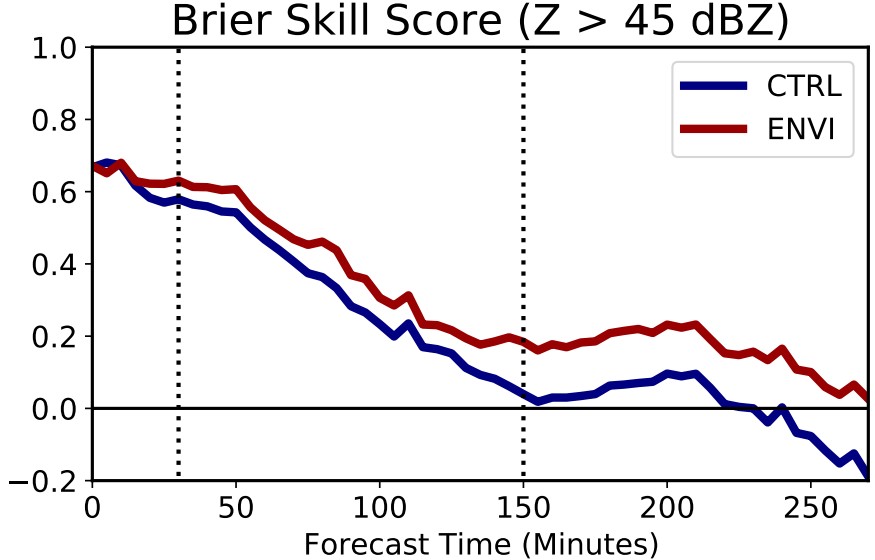

**Figure 15.** The BSS evaluating the P($Z$ > 45 dBZ). Bold blue and red lines correspond with CTRL and ENVI, respectively. The horizontal black line marks the line of no skill. Vertical dashed lines mark the times when forecast probabilities are evaluated in Figure 14.

and environmental observations (surface and sounding) are consequently more skilled; the QLCS moves faster and has larger
updrafts.

The data assimilation experiments conducted in this study are relatively simplistic in their treatment of observations. Environmental observations are interpolated from model output with gaussian noise to simulate observation error. This strategy is likely adequate for in-situ observations that directly observe the atmosphere (e.g., soundings), but underrepresents retrieved profile errors for remote sensing systems (e.g., doppler wind lidars, atmospheric emitted radiance inferometers). To better
understand the impact of boundary layer profiling systems in future data assimilation experiments, it is imperative to more accurately represent instrument errors. Further, high-resolution modeling studies of the simulated environment are needed to quantify representativeness errors that also introduce uncertainties during data assimilation (e.g., Janjić and Cohn, 2006; Hodyss and Nichols, 2015).

Future work using this OSSE configuration will assimilate different simulated boundary layer profiling instruments to un-
derstand their impact on convective initiation and QLCS evolution. Assimilating thermodynamic and kinematic profiles of the boundary layer can improve representation of the lower troposphere and consequently increase convective initiation forecast skill (e.g., Coniglio et al., 2019; Degelia et al., 2019, 2020; Chipilski et al., 2020). These OSSEs will also help determine an optimal assimilation strategy including: sampling frequency and spatial density. Insight gained will help determine how these instruments should be deployed during future field campaigns to study and understand high-impact weather in the southeast
U.S.



Finally, future studies should develop techniques to quantify the expected level of maximum forecast skill based upon event predictability. Understanding the limits of practical predictability (e.g., Melhauser and Zhang, 2012; Zhang et al., 2015; Flora et al., 2018) should help focus efforts by users looking to optimize the design of, and the assimilation of, observations from new observing networks in the coming decade.

*Code and data availability.* The software and scripts used to generate the initial ensemble forecast, conduct data assimilation, run the free forecast ensemble, and post-process model output can be found accessed at https://doi.org/10.5281/zenodo.7109050. In lieu of recreating the data, initial conditions for the nature run, assimilated observations, and the initial prior ensemble at the time of the first assimilation cycle can be accessed at: https://doi.org/10.5281/zenodo.7126769

*Author contributions.* Jonathan Labriola created the experiment framework and conducted the analysis. Louis Wicker provided the funding,
project administration, and assisted in the project creation and evaluation.

*Competing interests.* The authors declare that they have no conflict of interest.

*Acknowledgements.* This research was performed while the author held an NRC Research Associateship award at the National Severe Storms Laboratory. Additional funding was provided by the Verification of the Origins of Rotation in Tornadoes Experiment-Southeast-United States of America (VORTEX-SE[USA]) project. The authors thank Keith Sherburn and Matthew Parker, who provided the initial sounding and
model settings for the QLCS case. Derek Stratman reviewed the journal article prior to submission and provided valuable feedback. Valuable local computing assistance was provided by Gerry Creager, Jesse Butler, and Jeff Horn. Soundings and CAPE calculations are performed with metpy software.



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
