# Peer review of "A method for generating a quasi-linear convective system suitable for observing system simulation experiments"

_EGUsphere, 2022_

## Author Comment (AC1)

**Reply to Reviewer 1**

This paper presents a new setup for an OSSE assimilation system. The main novelties are the nature run (front convection) and the ensemble generation. I like very much the choice of the nature run but I have some questions regarding the initial ensemble. The ensemble is too far from the nature run (see below). I also think that adding results of the free ensemble (no assimilation) would make the results more interesting. For that reason, I recommend major revision.

We thank the Reviewer for their helpful feedback. We tried to incorporate as many suggested changes as possible and as a result believe the manuscript is improved. If a change was not enacted fully, we justify our position in the reply.

1)   *The initial ensemble seems quite far from the nature run as it is shown in Figure 7, where the nature run profiles are often far from the ensemble spread. I think this is a result of the generation ensemble, the ensemble which has run for over 24 h (with full physics) in comparison with the nature run. These are difficult conditions for an assimilation system, and I do not think that these are quite realistic (the errors in the cold front are ~4K). I wonder how the nature run would be if the same procedure would have been applied (run each part of the front independently for 24 hours). I guess that to have a bad ensemble is the author's choice, but this should be clear in the abstract or introduction: not only that the errors are non-gaussian, but also that the nature run profile is often out of the ensemble spread with the consequence of a much weaker convection in all ensemble members.*

Running the forecast simulations for 24 hours causes wind profiles near the surface to slow down because there is no pressure-gradient force to counteract friction. This provides an opportunity to assess whether assimilated environmental observations can improve the wind profile. If we generate a nature run simulation using the same procedure, the QLCS would be much weaker because there is little wind convergence along the frontal boundary. Because a primary goal of this study is to create a tornadic QLCS, we plan to keep the current nature run configuration.

We agree that it is important to highlight that this study introduces a methodology to generate an initial ensemble that is degraded. We added the following to the abstract on lines 13–14: "*This purposely degraded initial forecast ensemble provides an opportunity to assess whether assimilated environmental observations can improve, e.g., the wind profile.*" We also added the following to the introduction on lines 86–89: "*These techniques lead to a purposely degraded initial ensemble that contains robust non-gaussian errors, such that the profiles from the nature run are often out of the ensemble spread. This ensemble of simulations allows for evaluation of the data assimilation system described herein.*" It is also discussed elsewhere, e.g. conclusions.

2)   *In figures 11, 12, 13, 14 and 15 would it be very useful to plot the skill of the initial ensemble without data assimilation. We could then see the skill of the assimilation system. In figure 13 you could also plot the nature run.*

We re-ran the forecast ensemble without data assimilation (we call this the "NoDA" case). We added the NoDA curves to our now Figures 11–16 per the Reviewer's request. We opted against including the nature run in Figure 13 because the resolution differences do not allow for a fair comparison of updraft volume characteristics (not apples-to-apples).

*3)   It would be interesting to see how realistic the prediction of reflectivity is. I suggest plotting the reflectivity of one member (without assimilation, CTRL and ENVI) at assimilation time and after a 3h forecast.*

We agree with the Reviewer and have added the requested figure (new Figure 15) using member 25. We plot the reflectivity at the lowest model level for the nature run and forecasts at 30 minutes and 150 minutes to match Figure 14.

*4)   The generation of turbulence in the nature run is not clear. I do not understand why radiation and surface friction are switched off, when these are the main drivers for boundary-layer turbulence. I also do not understand why you want to run the simulation for 12 hours, which is much longer than any boundary layer process*

We turned off radiation and surface friction to ensure that the median profile of the initial environment remained unchanged, and to avoid initiating spurious convection, which would alter turbulent processes. While 12 hours is longer than most BL processes, we need to simulate several eddy-turnover cycles to reach a kind of quasi-statistical equilibrium, such that the generated structures are robust and representative of a realistic boundary layer under the considered conditions.

Minor:

*1)   Equation 1. Set the appropriate limits. The equation is only true if the argument is larger than 0.*

We politely disagree with the need for limits to be specified and/or the limits suggested in the comment. The manuscript notes that the impact of the perturbation decreases with distance from the western and bottom domain bounds. In Eq. 1 mentioned here, cosine can accept 0 exactly, which happens at the western and bottom bounds, and that case maximizes the perturbation since $\cos(0)=1$. While we could say that x, y, z should be positive, that's not technically a requirement either based on the formulation of ß and the fact that cosine is an even function.

*2)   Equations 1,2,3: the letter f is used for two different functions. This is confusing.*

We agree with the reviewer and have changed $f(x)'$ to $f(y)$ to $f_p(x)$ and $f_w(y)$, respectively, to denote that the functions relate to perturbations and waves.

*3)   Please specify if the simulations to generate the ensemble are done with the nature-run physics or with the forecast physics.*

We now make this clear (ensemble physics) in the Section 3 intro and on line 217.

*4)   Please specify which cloud cover scheme (if any) is used in the nature run and forecast physics. Also specify if the turbulent schemes are 1D or 3D.*

We now clarify these in the text (no cloud scheme, nature run is 3D, coarser ensemble uses 1D).

5)    *Line 265: I guess there is a not missing.*

Fixed.

6)    *I cannot see the soundings in Figure 10.*

We amended the figure to improve legibility.

7)    *Do you see much updraft helicity in the forecast?*

We chose not to focus on this area of analysis in the current work, but may do so in a future study.

8)    *OSSE assimilation systems can also be used for improving the assimilation system independently of the errors in the model physics (see for example Zeng et al. 249, 105282, 2021,https://doi.org/10.1016/j.atmosres.2020.105282)*

We agree with the Reviewer and have added mention to this point on line 37.

9) *Figure 11. Innovation is probably not the right label for the y-axis.*

We updated Fig. 11 to state root mean square error based on the recommendation of Reviewer 2. We agree RMSE is a more appropriate phrase since we are not comparing the prior/posterior against simulated observations.

---

## Author Comment (AC2)

**Reviewer Comments 2**

*In this manuscript, the technique used to generate a nature run that is representative of a tornado outbreak in the southeastern United States is introduced. Since past studies conduct OSSEs to simulate evolution of supercellular convection by generating a "warm bubble" into an unstable and highly sheared environment, it is meaningful to perform idealized OSSE that simulates the evolution of a convective line initiated via a frontal boundary in a highly-sheared and modestly-unstable environment. Creating OSSEs that simulate different storm modes and environments can help better understand how assimilated observations impact the environment and the subsequent evolution of convection. Forecasts that assimilate radar and environmental observations are found to be more skillful than assimilating radar data only. Environmental observations help correct wind profile error and increase convergence. The authors introduce a new method to create initial ensembles, however, it is not addressed very clearly. I recommend accepting this paper after a minor revision.*

We thank the Reviewer for their helpful feedback. We tried to incorporate as many suggested changes as possible and as a result believe the manuscript is improved. If a change was not enacted fully, we justify our position in the reply.

*The first question is about the nature run. It is not very clear what the final setup of the nature run is. It looks to me like the nature run is initialized from the environmental sounding shown in Fig. 1a. A frontal boundary is added to provide a mechanical forcing for convection initialization. A turbulence simulation is conducted to help introduce more realistic eddies. Then the perturbations of the u,v,w, qv, and theta fields from the 12-h forecasts of the turbulence simulation are added back to the initial condition of the nature run. I think it is better to specify more clearly what the final setup of nature run is before section 2.4, similar to the setup descriptions (step 1 to 4) in section 3.*

We agree with the Reviewer that this was confusing. We added a list at the start of Section 2 as in Section 3. We also swapped Section 2.2 and 2.3 to better match the order of operations.

*It is unclear to me how the 40 initial ensembles are generated. It is mentioned in 200, cold and warm sector simulations for each ensemble member are assigned a land surface type. How many warm and cold sector simulations are conducted? Based on the captions in Fig. 7, it looks to me there is only one warm sector simulation and one cold sector simulation (the ones that use the unperturbed sounding). Are the 24-h forecasts from cold and warm sector simulations blended together in different ways (with different times and locations) to form different initial cold front boundary for 40 different ensemble members? In line 360, it is mentioned that each forecast member is initialized from the same sounding. Is it the same sounding as the nature run? What are the perturbed soundings in Fig. 7 used for? Are there 40 perturbed soundings in warm sectors and another 40 ones in cold sectors? It is better to summarize the setup descriptions of the initial ensembles before section 3.3 instead at the beginning of the section 3. Summarize what are the difference in the 40 initial ensembles (e.g., do they use the same or different sounding, cold front boundary, land surface type, and potential temperature perturbations, etc., for simulation?). The timeline is also not very clear to me. What is the time setup for nature and the runs to generate initial ensembles? Fig. 9 only shows the time setup after generation of the initial ensembles.*

We understand that this section is verbose and appreciate the Reviewer's patience. First, we believe summarizing the steps is best left at the start of the section because it directs the reader in plain language as they progress through the subsections. We believe the questions raised here are addressed in the Section 3 intro list and in Section 3.1. We will review the procedure here based on the text and hope that it now makes sense.

- For each of the 40 members, a cold-sector and warm-sector simulation are initialized.
  - The warm-sector initial sounding is from the nature run initial sounding.
  - The cold-sector initial sounding is made by applying perturbations to the warm-sector initial sounding.
- Each member is assigned a land-use category (see Table 1), which are identical for the cold- and warm-sector simulations (except for open-water bodies—warm sector only).
- Additionally, random perturbations of potential temperature are added.
- For each member, the cold- and warm-sector simulations are integrated forward 24 hours using the ensemble model physics settings.
- Figure 7 shows the domain-averaged profiles of wind and temperature at the conclusion of those simulations (thin lines). The bold lines are the soundings used to initialize each of the cold- and warm-sector simulations.
- The cold- and warm-sector simulations are then blended together using the described weighting function for each member to arrive at the initial conditions used to create the ensemble.

*Line 235: should be "Due to the idealized nature".*

Fixed.

*Usually for OSSEs, model forecasts are verified against the true state instead of the observations due to the errors of the observations. Did you conduct verification using observations or the "true" state from nature run? If using the latter, RMSE instead of RMSI should be used.*

We updated Fig. 11 to state root mean square error.. We agree RMSE is a more appropriate phrase since we are not comparing the prior/posterior against simulated observations.

*It seems assimilating conventional observations together with radar data produces much stronger updraft relative to assimilating radar data alone. Environmental observations help correct wind profile error and increase convergence. Which one do you think has more influence on the analysis of the updraft, sounding, or surface observations?*

We believe that radar observations have the largest impact on the initial updraft intensity. After, the assimilated sounding observations enhance convergence and thus help maintain updraft intensity.